# Assessing Attitudes and Participation Regarding a Pilot COVID-19 Workplace Vaccination Program in Southern Germany Considering the Occupational Health Perspective—A Mixed Methods Study

**DOI:** 10.3390/vaccines11061082

**Published:** 2023-06-09

**Authors:** Anke Wagner, Kamil Keles, Christine Preiser, Anna T. Neunhöffer, Jana Soeder, Juliane Schwille-Kiuntke, Monika A. Rieger, Esther Rind

**Affiliations:** Institute of Occupational and Social Medicine and Health Services Research, University Hospital Tübingen, Wilhelmstraße 27, 72074 Tübingen, Germany

**Keywords:** COVID-19 vaccination, COVID-19 vaccination campaign, vaccination in workplace settings, occupational medicine, occupational health services, employees, mixed methods research, qualitative research, quantitative research, Germany

## Abstract

This mixed methods study retrospectively assessed attitudes and participation of employees, occupational health personnel, and key personnel regarding the rollout of a pilot COVID-19 workplace vaccination program in five German companies in May/June 2021 in Baden-Württemberg (Southern Germany) by combining survey data and qualitative interviews. A total of 652 employees completed a standardized questionnaire and we conducted ten interviews with occupational health personnel and key personnel with other professional backgrounds organizing the pilot workplace vaccination program. Survey data were analyzed descriptively and interviews were audio recorded, transcribed verbatim, and analyzed using qualitative content analysis. Employees participated widely in COVID-19 vaccinations at their workplaces, and most employees (n = 608; 93.8%) had a full COVID-19 immunization at the time of the survey. The main advantages of the pilot COVID-19 workplace vaccination program were seen in the flexible and time-saving vaccination offer as well as the trust in and long-standing relationship with occupational health physicians. The main disadvantage of the pilot vaccination offer was increased workload for occupational health personnel, especially during the roll-out phase of the program. The pilot COVID-19 workplace vaccination program was predominantly positively assessed, and the important role of occupational health services in managing the COVID-19 pandemic was highlighted. The main criticisms of the COVID-19 workplace vaccination program related to the high organizational and administrative burden. Findings from our study can support the development of future programs for the administration of generally recommended vaccination in the workplace setting in Germany.

## 1. Introduction

In late 2019 and early 2020, the severe acute respiratory syndrome Coronavirus 2 (SARS-CoV-2) spread worldwide, resulting in high numbers of infection rates, hospitalizations, and number of deaths [1]. On 11 March 2020, the World Health Organization (WHO) declared the spread of the coronavirus disease (COVID-19) a pandemic [1]. Since 2020, different public health attempts have been made to reduce rapid transmission through physical or social distancing measures [2]. Other measures included maintaining personal hygiene (especially washing hands, coughing and sneezing etiquette), wearing mouth and nose protection, and performing regular SARS-CoV-2 antigen testing [3].

Simultaneously, the development of suitable vaccines against COVID-19 was intensified, and at the end of 2020, the first vaccine (Comirnaty^®^ developed by BioNTech, (Mainz, Germany) and Pfizer (New York City, NY, USA)) was approved in the European Union by the European Medicines Agency (EMA) [4]. In 2021, three additional vaccines received conditional marketing approval by the EMA: Jcovden^®^ (previously COVID-19 Vaccine Janssen (Beerse, Belgium)), Spikevax^®^ (developed by Moderna (Cambridge, MA, USA)), and Vaxzevria^®^ (developed by AstraZeneca (Oxford, UK)) [4]. These vaccines proved to be very effective in reducing infection rates, illness severity, hospitalization, and mortality in the fight against SARS-CoV-2 [5,6].

In Germany, the national COVID-19 vaccination campaign started in late 2020 and early 2021. In phase one, only high-risk groups and particularly exposed individuals, such as healthcare workers, received COVID-19 vaccination due to a justified vaccination prioritization [7]. Primary COVID-19 vaccination sites were special vaccination centers established for this purpose and mobile vaccination teams [7]. In phase two, general practitioners in their practices (April 2021) and occupational health physicians (June 2021) were included in the national vaccination campaign [7,8]. Beginning June 2021, there was no longer any vaccination prioritization, and every resident in Germany older than 12 years could be vaccinated against COVID-19 [7]. The inclusion of occupational health physicians in the vaccination campaign had two effects: offering COVID-19 vaccinations in companies added a new setting to the vaccination campaign, and a group of physicians was included in the vaccination campaign who usually only administer vaccines based on occupational health indications and not due to general recommendations (i.e., public health perspective).

To prepare for the planned inclusion of workplace settings in the national COVID-19 vaccination campaign, within the framework of pilot projects, occupational health physicians in Germany had already been allowed to carry out COVID-19 vaccinations in the workplace since the beginning of April 2021 [8]. Occupational health physicians had previous experience with vaccinations in the workplace (based on occupational health indications and—in some companies—against influenza), and were therefore able to provide valuable support in this pandemic situation [9]. Therefore, those pilot vaccination programs in the workplace were initiated by policy makers and implemented during the third COVID-19 wave, which occurred in Germany during spring 2021, with high infection rates among the German population [10] and a shortage of vaccines. In these pilot projects regarding workplace vaccination programs, employees received an offer for COVID-19 vaccination at their company despite the current vaccination prioritization, i.e., at an earlier timepoint than based on their individual situation or health condition [8]. The first COVID-19 pilot workplace vaccination programs were limited to Baden-Württemberg, Bavaria, Lower Saxony, and Saxony [11,12,13,14] as, due to the shortage of vaccines, no further COVID-19 workplace vaccination programs were planned in other federal states of Germany [11].

Within the pilot project, in Baden-Württemberg, COVID-19 workplace vaccination programs were first carried out in 15 selected companies [12]. These 15 companies were part of what was defined as critical infrastructure [12], for example, companies involved in producing medical products and devices, food, and energy [15]. The selection of the 15 companies was also based on their size so that COVID-19 vaccinations were performed in both larger but also in three small-sized companies with fewer employees [12]. From the start of the vaccination campaign in the pilot companies in May until June 2021, more than 12,000 employees in Baden-Württemberg have been vaccinated in these companies by occupational health physicians, and initial experiences have been evaluated for use in the ongoing development of the national COVID-19 vaccination strategy [12]. However, the implementation of the pilot projects in these companies in Baden-Württemberg was not systematically accompanied by researchers. To date, no studies have been conducted to evaluate the implementation and acceptance of the COVID-19 workplace program in the workplace setting considering multiple perspectives, such as those of providers, employers, and employees.

The study presented here focused on this issue, and was performed in the context of the pilot COVID-19 workplace vaccination program, considering the perspectives of occupational health personnel and persons closely involved in the rollout of the program, as well as employees in the above-mentioned companies in Baden-Württemberg, Germany. For this publication, we focused on the following two research questions:What were the attitudes of employees, occupational health personnel, and key personnel closely involved in the pilot COVID-19 workplace vaccination program toward COVID-19 workplace vaccinations?What was the level of employee participation in the program, and how was the pilot COVID-19 workplace vaccination program retrospectively assessed by occupational health personnel and other key personnel closely involved in the program?

By presenting the perspectives of employees, occupational health personnel, and other key personnel, conclusions can be derived about experiences with and the general acceptance of COVID-19 vaccinations in the workplace setting. We expect the findings to highlight companies’ resources and challenges regarding the organization and management of occupational health services in exceptional circumstances, such as the COVID-19 pandemic. In Germany, there is so far little experience and few studies on generally recommended vaccinations by occupational health physicians in the workplace. It is therefore important to consider this type of COVID-19 vaccination in addition to COVID-19 vaccinations by general practitioners or COVID-19 vaccinations in vaccination centers. Therefore, for key decision makers, the results may offer first indications for the development of future occupational vaccination programs with regard to generally recommended vaccinations, their quality, and their acceptance within the companies.

## 2. Materials and Methods

### 2.1. Study Design

In agreement with the Ministry of Social Affairs, Health and Integration in Baden-Württemberg, Germany, we performed a mixed methods study that retrospectively evaluated the pilot COVID-19 workplace vaccination program in different companies in the German federal state Baden-Württemberg. “Mixed methods research is an approach that combines the strengths of quantitative and qualitative research for the purpose of obtaining a richer and deeper understanding” [16]. “Mixed methods also is well suited for interdisciplinary research (…)” [17]. The mixed methods study is part of a larger explorative study project investigating how SARS-CoV-2 affected working conditions in different enterprises and workplaces since the beginning of the pandemic in Germany, and which implications can be derived for maintaining safe working conditions in exceptional circumstances [18]. The initial results of the main study are published elsewhere [19,20,21].

This mixed methods study comprised standardized surveys with employees and qualitative interviews with members of crisis teams, occupational health personnel, as well as key personnel with activities related to sustainability, occupational health management and process management closely involved in the organization and rollout of the COVID-19 workplace program. Among the variety of mixed methods designs, we opted for an embedded design according to Edmonds and Kennedy (2017) [22], in which one type of data is more relevant to the research team than the other(s). This is a common approach in mixed methods studies in health services research [23]. Due to the explorative and retrospective character of the overall study project and the limited time for data collection, we decided in favor of an unequal weighting with emphasis on qualitative methods. During interpretation, the qualitative and quantitative findings complemented each other. Thus, we gained a more differentiated description and comprehensive understanding of the multi-level perspectives regarding attitudes toward and participation in COVID-19 workplace vaccination programs.

We followed the guidelines for Good Reporting of A Mixed Methods Study (GRAMMS) reported by O’Cathain and colleagues [24]. We also considered the mixing procedure of integration described by Zhang et al. by collecting and analyzing the qualitative and quantitative data independently from each other [16].

### 2.2. Recruitment and Study Setting

The Ministry of Social Affairs, Health and Integration in Baden-Württemberg supported the study and, in agreement with the 15 participating companies, provided a list of contact persons that E.R. contacted by email in August 2021 to inform the stakeholders within the companies about the planned retrospective evaluation. Five companies agreed to participate and cooperation agreements were developed and signed. Two of the five companies consented to take part in the employee survey, and four of the five companies agreed to participate in qualitative interviews. The five companies cover medical technology, medical device production, pharmaceutical production, supply chains for food, medicines, water, energy, and other essentials, and energy supply. The remaining ten companies did not agree to participate. Reasons for non-participation were mostly related to a lack of time for staff to participate in the employee survey or an interview.

The timing of the study period, October 2021 to January 2022, coincided almost simultaneously with the fourth wave of COVID-19 in Germany (winter 2021), with high infection rates in the German population due to the contagious delta variant of SARS-CoV-2 [10], and the beginning of the “booster campaign” for repeat vaccination.

### 2.3. Employee Survey

To encourage participation in the employee survey, the study team organized an additional informational event for employees at each company prior to data collection. Two companies with sectors in medical technology and supply chains for food, medicines, water, energy, and other essentials agreed to participate in the employee survey. Data collection took place between November and December 2021. Employees had the option of completing the questionnaire either online using the established survey tool Unipark [25] or using paper and pencil.

The development of the employee survey is described in detail elsewhere [18,19,20]. In short, the questionnaire includes self-developed items and questions from previous studies [20,26,27] covering the following topics:I.Individual and sociodemographic aspects (e.g., age), and workplace characteristics (e.g., professional activity, performing shift work);II.Perception of SARS-CoV-2 in general and the impact of COVID-19 on the personal environment;III.Information relating to COVID-19 vaccination in general and COVID-19 vaccination in companies;IV.Attitudes toward health and safety measures to prevent SARS-CoV-2 infections in the workplace environment.

For this study, specific questions related to COVID-19 vaccination in companies are presented in greater detail. As there were no existing questionnaires that captured attitudes toward COVID-19 vaccination in companies, the following self-developed questions were constructed:One item asking about the possibility of being vaccinated against COVID-19 by occupational health physicians (“In general, how would you rate the possibility of being vaccinated against COVID-19 by your occupational health physician?”) on a Likert scale from 1 (=“very positive”) to 5 (=“negative”).One mean score on aspects related to the organization of the pilot COVID-19 vaccinations within the company (e.g., satisfaction with organization, provided information, waiting time, medical education, and availability of a contact person) ranging from 1 (=“not satisfying”) to 5 (=“very satisfying”). Cronbach’s alpha for this score was satisfying with 0.86.One item addressing possible reasons (multiple answers and free text answer) why the COVID-19 vaccination was not received in the workplace (“What were the reasons you did not receive the vaccination at work?”).One item addressing possible reasons (multiple answers and free text answer) for receiving the COVID-19 vaccination at the workplace (“What were the most important reasons for you to receive the vaccination at your company?”).Two items regarding participation of employees in the pilot COVID-19 workplace program: one item specifying between the first and second COVID-19 vaccinations (“Which of the vaccination(s) against COVID-19 did you receive at work?”), and one item asking where the COVID-19 vaccination was administered (“Where did you get vaccinated?”).One item addressing COVID-19 vaccination status (“Have you already received a vaccination against COVID-19?”), and, if the person indicated that they had been vaccinated, one item asking for the specific COVID-19 vaccine (“Which vaccine did you receive?”).

Overall, 652 employees from two companies participated in the survey. As shown in Table 1, more than half of the employees were male (n = 352; 54.7%) and on average 42.8 years (SD = 12.2) old. The majority of employees had no supervisor function (n = 547; 84.9%) and were working full-time (n = 522; 81.2%). Most of the employees did not work in shifts (n = 548; 85.6%) and had a long-term contract (n = 618; 96.3%). The average professional experience within the study population was 18.1 years (SD = 11.9).

### 2.4. Qualitative Interviews with Occupational Health Personnel and Other Persons

In autumn 2021, A.W. contacted the companies that agreed to participate in the qualitative part of the study to arrange interview appointments with eligible participants. Inclusion criteria were as follows: participants had to be part of the occupational health service and/or closely involved in the management of the pilot COVID-19 workplace vaccination program. The participants were informed about the study and signed a declaration of consent. In addition, participants could decide whether the interview should take place by telephone or as a videoconference.

The interview guide was developed mainly by A.W. with feedback from researchers and members of the Institute of Occupational and Social Medicine and Health Services Research at the University Hospital Tübingen with backgrounds in occupational medicine, internal medicine, general medicine, and health services research. The interview guide included specific topics about the pilot COVID-19 workplace vaccination program (please see Appendix A) and focused especially on the following:Identified advantages and disadvantages of COVID-19 vaccinations in companies;Perceived response of employees to the pilot COVID-19 vaccination offer;General assessment of the pilot COVID-19 workplace vaccination program.

From October 2021 to January 2022, A.W. conducted nine interviews. K.K. carried out another interview under the supervision of A.W. so that ten interviews were the basis for further analysis. Nine interviews were conducted via videoconference and one via telephone. The interviews lasted 31 min on average, varying in duration between 25 and 46 min. After each interview, a short questionnaire was completed that included central socio-demographic information of the interview partners and space to add further comments on the perceived atmosphere of the interview. Five occupational health physicians and five persons with different professional backgrounds responsible for organizing the pilot COVID-19 workplace vaccination program took part in the interviews. The interview partners were on average 49.5 years old, and occupational health physicians in our sample had an average professional experience in occupational medicine of 14.6 years. An overview of the study population is shown in Table 2.

### 2.5. Data Analysis

Survey: The data analyses comprised mainly descriptive analyses using IBM SPSS^®^ Statistic^®^ Version 28 (IBM Corp., Armonk, NY, USA). For the following publication, we focused on data regarding the topic “COVID-19 vaccination in companies” and the included attitudes of the employees. We calculated frequencies of the answers to each question. No data were imputed.

Qualitative interviews: All interviews were audio recorded and transcribed verbatim by a professional transcription agency using a simplified transcription system [28]. A.W. and K.K. checked the transcripts carefully to make sure no information was lost. A.W. then pseudonymized the interviews, with all names and places replaced. MAXQDA 2020 (VERBI Software, Berlin, Germany) was used for organizing the data [29]. The data analysis followed the main steps according to qualitative content analysis by Mayring [30] and was performed jointly by A.W. and K.K. to ensure quality assurance. The main steps of the data analysis included the development of a category system using both a deductive (derivation of content of the semi-structured interview guide) and inductive (identifying additional themes in the material) approach. To conduct the coding of the interview passages in a consistent manner, a coding guide was developed and discussed by A.W. and K.K. The final category system included six main categories and various subcategories. An overview of the category system is presented in Appendix A. A.W. and K.K. carried out a more in-depth analysis of each category and subcategory based on the final coded text passages from the interviews. All results of the in-depth analysis performed were discussed in two separate meetings with two to four other researchers from the Institute of Occupational and Social Medicine and Health Services Research. Following the analysis, all quotations included in this study were translated from German into English in cooperation with a native speaker. As we did not apply a conversation analysis approach [31], and focused more on the overall content and meaning of the data collected, we do not expect any significant loss of meaning due to the performed translation. 

For this publication, we focused on categories that (1) reflect attitudes of the study population toward COVID-19 vaccinations in the workplace and that (2) provide an assessment of the pilot COVID-19 vaccination workplace program in companies. 

The concept of attitude is a well-researched construct in social psychology with affective, cognitive, and behavioral components [32]. Attitudes contain, in our study, the included employees’ direct opinions on COVID-19 vaccinations in their workplaces, as well as the identified advantages and disadvantages of COVID-19 vaccinations in companies, which were comprehensively discussed during the qualitative interviews. Assessment, on the other hand, can be defined as “an opinion or a judgment about someone or something that has been thought about very carefully”. [33] Assessment included in our survey the recorded participation rate in the pilot COVID-19 workplace program, as well as the COVID-19 vaccination status of employees. In the qualitative interviews, topics with an assignment to the category of assessment were primarily statements on the perceived response and a general retrospective appraisal of the pilot COVID-19 workplace vaccination program.

### 2.6. Ethical Approval

The study was approved by the responsible local ethics committee of the Medical Faculty, University of Tübingen and University Hospital Tübingen (No. 423/2020BO). Only study participants who consented to anonymous processing of their data were included in the quantitative and qualitative analyses of the data obtained. 

## 3. Results

Our mixed methods study comprised findings from standardized employee surveys, as well as qualitative interviews with occupational health personnel and other key personnel involved in the pilot workplace vaccination program. Based on the research questions, we assigned our results to the following main topics:Attitudes toward COVID-19 vaccinations in workplaces;Assessment of the pilot COVID-19 vaccination workplace program in companies.

### 3.1. Attitudes toward COVID-19 Vaccinations in Workplaces

We observed positive attitudes from employees regarding COVID-19 vaccinations in companies. The employees rated the possibility of being vaccinated by occupational health physicians very positively, with a mean of 1.29 (SD = 0.7) (score ranging from “1 = very positive” to “5 = negative”). Different aspects of organization, provided information, waiting time, medical education, and availability of a contact person during workplace vaccination were also evaluated very positively by the employees with a mean of 4.70 (SD = 0.6) (score ranging from “1 = not satisfying” to “5 = very satisfying”). 

Within the survey, we also assessed possible reasons for and against carrying out COVID-19 vaccinations in the workplace. In most cases, the reason given for not receiving the two COVID-19 vaccinations in the company was having an earlier vaccination appointment elsewhere (n = 194; 29.8%). Only few employees desired a separation of work when using health services (n = 7, 1.1%) or had a lack of trust in occupational health physicians (n = 4, 0.6%). The most frequently mentioned reasons for receiving the COVID-19 vaccination in the company were fast appointments (n = 325, 49.8%), the easily accessible character of the vaccination offer (n = 241, 37.0%), and that receiving the vaccination was possible during working hours (n = 171, 26.2%). 

The results of the survey corresponded with findings from the qualitative interviews, in which the possible advantages of COVID-19 vaccination in companies were discussed in more depth. The interview partners reported several examples of general advantages of COVID-19 workplace vaccinations. Some advantages were also closely related to the nature of the offer and included the following aspects: easily accessible, available on-site and during working hours, as well as time-saving (no waiting time and no appointment necessary).


*Occupational health physician, medical technology (Interview 5): “We are directly on site. You don’t have to walk long distances or wait for a long time, but we are directly where the employees are (…).”*


Furthermore, the vaccination offer was targeted to various groups of people who are difficult to reach in other settings, such as general practices.


*Occupational health physician, energy supply (Interview 1): “And that many people who perhaps wouldn’t go to the physician at all, simply drop in at the occupational health physician’s office during working hours to get vaccinated, but don’t have to take half a day off work to get an appointment at the general physician’s and then sit in the practice room for two hours.”*


Other identified advantages encouraging positive attitudes were related to the occupational health physicians themselves who (a) were experienced in dealing with vaccinations due to previous vaccination campaigns (for example influenza vaccinations), (b) reduced the workload of other healthcare services, and (c) made a significant contribution to supporting the national German COVID-19 vaccination campaign.


*Occupational health physician, pharmaceutical production (Interview 10): “So (…), there are simply more physicians who are vaccinating and supporting the vaccination campaign, that’s for sure.”*


Other advantages were that occupational health physicians were well-known by the employees, had established a long-lasting trusting relationship with employees due to years of occupational medical care, and knew exactly what medical information individual employees needed.


*Manager for sustainability and social responsibility, medical technology (Interview 2): “Simply the physical proximity, they [occupational health physicians] know the staff, they also know the high-risk patients, they know where they may or may not need to do a little more explaining. (…) The employees, in turn, know the occupational health physician. It’s a completely different relationship of trust than when I go to an anonymous vaccination center.”*


Another advantage included the ability of occupational health physicians to suspend certain non-priority tasks for a period of time, which allowed them to take time for COVID-19 vaccinations and related activities, such as counselling and education.


*Occupational health physician, different companies (Interview 4): “(…) We can put our preventive examinations aside and can then face the current issues that arise (…).”*


Moreover, the interview partners identified advantages for the company. The offered COVID-19 vaccination was seen by employees as a sign of appreciation from the company and can therefore increase employee satisfaction and retention in the future.


*Head of staff department, energy supply (Interview 6): “(…) And of course it is an advantage (…) for the employees when they say: The company does something for us, and the company applies for such an initiative. And then, of course, at the end of the day, it has to do with employee satisfaction, with employee loyalty, with all kinds of other factors (…).”*


We also discussed possible disadvantages of COVID-19 workplace vaccinations. In three interviews, occupational health personnel thought that there were no disadvantages at all for providing COVID-19 vaccinations in companies. According to the research participants, further possible disadvantages were found on the side of the companies. COVID-19 vaccinations generated high costs for the respective companies, and may have also resulted in envy from other companies who could not offer these vaccinations to their employees.


*Head of staff department, energy supply (Interview 6): “(…) There may be a kind of jealousy factor, it must be said, because of course we organized all this during working hours (…)”*


A substantial disadvantage of the COVID-19 vaccinations was perceived by the occupational health physicians themselves. First, occupational health physicians indicated they did not receive financial reimbursement for COVID-19 vaccinations. Another disadvantage was the increased workload for occupational health physicians, especially during the COVID-19 vaccination program. This manifested itself in additional workload, lack of time for other work activities due to high work intensity, and other work activities not being completed. Occupational health physicians also mentioned that they have to deal with the consequences of COVID-19 vaccinations, such as vaccination reactions and potential side effects. 


*Occupational health physician, medical technology (Interview 5): “So, the employees arrived in swarms afterwards with their, let’s say, vaccination side effects. Yes, who performed the ECG [electrocardiogram], who took the blood, who did the myocardial diagnostics or so. That was us (…).”*


### 3.2. Assessment of the Pilot COVID-19 Vaccination Workplace Program in Companies

The assessment of the pilot COVID-19 vaccination program in companies can be related to employee participation in the uptake of COVID-19 vaccinations in the workplace. A total of 376 employees (57.7%) received both the first and second vaccinations in their companies. In most cases, the COVID-19 vaccinations were received in the offices of occupational health physicians (n = 392, 60.1%) and vaccination centers (n = 161, 24.7%). Only few employees received their vaccinations from a general practitioner (n = 44, 6.7%) or from another medical specialist (n = 23, 3.5%). Most of the employees (n = 608, 93.8%) stated having full COVID-19 immunization at the time of the survey (November/December 2021). Only 28 employees (4.3%) reported not being vaccinated at all, and 8 employees (1.2%) had received only one dose of a COVID-19 vaccine at the time of the survey. Most employees (n = 372, 57.1%) were vaccinated with the Moderna mRNA vaccine (Spikevax^®^), followed by the BioNTech/Pfizer mRNA vaccine (Comirnaty^®^) (n = 215, 33.0%). Only a small percentage of employees were vaccinated with the Johnson & Johnson (Jcovden^®^ previously COVID-19 Vaccine Janssen) (n = 16, 2.5%) or AstraZeneca (Vaxzevria^®^) (n = 29, 4.4%) vaccines. 

Furthermore, the assessment of the pilot COVID-19 workplace vaccination program can be further strengthened with insights from the qualitative interviews regarding the perceived response to the pilot COVID-19 vaccination offer. Most employees expressed a high willingness to be vaccinated, which was also reflected in a rapid uptake of vaccination appointments.


*Practice assistant, medical device production (Interview 7): “And then we have the appointments—there were about 1100 appointments—they were then activated and I think within three hours they were all taken.”*


Likewise, there was a good atmosphere during the vaccination campaign and the vaccinated individuals reacted positively with feelings of gratitude, satisfaction, enthusiasm, and tears of joy.


*Manager for quality and sustainability, pharmaceutical production (Interview 8): “(…) And the fact that we said we’re now organizing this for everyone, was for the employees … that was a real relief. And also the atmosphere … we spent most of our time in home office because of Corona. Some colleagues didn’t see each other for months, and then we met in the vaccination center. We weren’t allowed to hug each other, but there were tears of joy, and we saw people again (…)”*


On the other hand, there was also a positive response on the side of the occupational health personnel who were united by the jointly conducted vaccination campaign and were also perceived differently by the employees.


*Occupational health physician, medical device production (Interview 9): “(…) And we are also perceived in a completely different way as an occupational health service, meaning as someone who helps in an emergency situation. (…) And that was suddenly a situation in which our occupational health service was perceived in a completely different way. We were much more visible for the other colleagues who normally don’t take up our services”*


From the point of view of the research participants, a pivotal reason for their positive response was related to the uncomplicated nature of the vaccination, which could be carried out during working hours.


*Team leader for occupational health management, medical technology (Interview 3): “(…) as already mentioned, […] that you can [go] during working hours, say I want to go over to the [vaccination] area, which is very close to the company, and can be vaccinated there. (…) So, yes … Only positive, actually…”*


In addition, some companies had the capacity to extend vaccination offers to family members and other companies.


*Head of staff department, energy supply (Interview 6): “(…) and it was not only our own employees with employment contracts at (…), but we also vaccinated external companies, we vaccinated affiliated companies, and we even vaccinated relatives (…).”*


However, in two companies there were some reports about employees showing a low level of willingness to be vaccinated.


*Occupational health physician, medical technology (Interview 5): “So we were completely surprised by the low level of willingness to be vaccinated. We had received 1200 doses of Moderna. We would not have thought that (…) the internal response was so low.”*


The occupational health personnel attempted to deal with the negative response by trying to increase vaccination uptake through various offers (during working hours, extensive information campaign).


*Occupational health physician, medical technology (Interview 5): (…) so they [employers] really went to incredible lengths to get the vaccine into peoples’ upper arms, but it’s proven difficult. There was a very big need at the beginning and suddenly there was vaccine left over. Yes, and that surprised and frustrated me, quite honestly.”*


In the interviews, the suspected reasons for negative responses were also discussed when offers and advertising for vaccinations were not successful. In addition, it was reported that employees with an immigrant background or a low level of education were particularly hard to reach.


*Manager for sustainability and social responsibility, medical technology (Interview 2): “It was simply difficult, yes. And we simply have a lot of employees who did not accept the offers. So certain … with an immigrant background mainly. The offer was poorly accepted, yes.”*


In addition to the perceived response of the employees, the interview partners themselves provided a final assessment of the pilot COVID-19 workplace vaccination program. Most of the interview partners rated it very positively. The main aspects of the positive assessment were related to the availability of vaccines within the COVID-19 workplace vaccination program at a time of vaccine shortage. These included, for example, the possibility of gaining experience in vaccinating larger groups of employees, the positive atmosphere during the program, and the opportunities to receive vaccine doses very quickly—and, therefore, to provide vaccination offers to employees and in some cases also to family members.


*Occupational health physician, energy supply (Interview 1): “(…) simply used it as an opportunity to obtain vaccines more quickly and in practically unlimited quantities. We received 2500 vaccine doses and were able to vaccinate practically everyone who wanted to be vaccinated, including family members.”*


Further aspects of the positive assessment of the pilot COVID-19 workplace vaccination program were related to the overall advantages for businesses, such as more exchange and cooperation between different companies, the possibility of easing some infection control measures in the company due to the vaccinated workforce, and reducing the risk of sick leave among employees. 


*Team leader for occupational health management, medical technology (Interview 3): “(…) That was definitely a benefit and, of course, it gives the company a sense of safety. That production can be maintained and that there are no major staff absences.”*


Finally, positive aspects were also attributed to the occupational health service, which experienced increased appreciation by company staff during the workplace vaccination program. Furthermore, occupational staff gained more routine experience in vaccinating over the course of the program.


*Occupational health physician, medical device production (Interview 9): “(…) So the fact that we vaccinate is widely recognized. This gives another boost to the occupational health service, that we also provide the vaccination service, that it is also associated with us, so that has already made a difference. And of course, we now have an extremely high level of routine in such procedures and organizational processes.”*


In contrast, there were also negative assessments regarding the pilot COVID-19 workplace vaccination program referring mostly to central challenges, such as high organizational and administrative burden and uncertainties during the program. The latter included, for example, the beginning and the duration of the program as well as the available number of vaccine doses.


*Practice assistant, medical device production (Interview 7): “The fact that it [the program] was constantly delayed or that at first we didn’t really know (…) on what kind of scale it would be? If we become part of the program, how many vaccine doses will we get? How many days are we talking about? But I don’t know… That was… There were a lot of things that came together.”*


Based on the assessment, the interview partners derived wishes and implications for further vaccination campaigns mainly related to better communication of information in different languages relevant and appropriate to the multicultural backgrounds of company staff. Other requests included the presence of a contact person to resolve common challenges and address uncertainties during the vaccination campaign.


*Manager for quality and sustainability, pharmaceutical production (Interview 8): “(…) Yes, and perhaps also provide contact persons who … who you don’t have to look for on the internet first, but who are somehow there to help and advise you.”*


## 4. Discussion

We conducted a mixed methods study to explore attitudes toward COVID-19 vaccinations in companies and toward a pilot COVID-19 workplace vaccination program from different perspectives. The mixed methods study comprised standardized surveys with 652 employees as well as qualitative interviews with 5 occupational health physicians and 5 other key personnel with different professional backgrounds. The results provide evidence for the broad acceptance of workplace vaccinations and can support the implementation of future programs for generally recommended vaccinations in workplaces as important settings for prevention.

### 4.1. Attitudes towards COVID-19 Vaccinations in Workplaces

The results of our mixed methods study revealed predominantly positive experiences and attitudes among employees and interview partners toward COVID-19 vaccinations in workplaces. The main reason for the positive assessment is the special nature of the vaccination offer with an easily accessible character. The interview partners addressed the flexible and time-saving vaccination offer on site and during work hours with short waiting times. Our results from the employee survey and the qualitative interviews correspond with results from other studies [34,35]. For example, a previous study from 2020, including 7494 employees from one German company, showed that good organization of the vaccination process and detailed information and education of the workforce play a major role in the decision to be vaccinated against influenza or COVID-19 in the workplace [35]. A previous intervention study to improve influenza vaccination rates in 2009 and 2010 showed that, in addition to the implementation of a specific information campaign, organizational factors such as simple and direct on-site vaccination also help to increase employees’ acceptance of vaccinations [36]. The nature of the conducted COVID-19 vaccination offer probably contributed to the high level of acceptance by the employees.

Other advantages cited in the interviews were that occupational health physicians were usually already known by the employees and had a long-standing relationship of trust. Furthermore, we identified that employees had a very positive view of the occupational health physicians’ work during the COVID-19 vaccinations. The long and trusting relationship with occupational health physicians was already reported in the German lidA (“leben in der Arbeit—life at work”) cohort study on work, age, health, and work participation, including a sample of 3039 employees [37]. The study showed that more than 62.1% of these employees were accompanied by occupational health physicians and more than 52.1% had contact with them in the last year [37]. The results of the lidA study do not indicate a shortage of occupational health physicians in Germany [37]. However, they suggest an unbalanced distribution of occupational health physician resources. Some occupational groups report a low level of contact with occupational health physicians, although more would be expected, while for others it is just the opposite [37]. This is supported by results of our previous study addressing the topic of workers’ health surveillance where many of the interviewed employees reported never having had personal contact to an occupational health physician [38]. An in-depth analysis of the current structure and quality of occupational healthcare in Germany is therefore recommended [37]. Another report on workplace COVID-19 vaccinations refers to this aspect. Occupational health physicians know the employees, have a professional relationship with them, and can fully inform them about the benefits and risks of vaccination [9], which is also reflected in our study results.

Besides the positive attitudes towards COVID-19 workplace vaccinations, disadvantages were also highlighted, including, for example, high costs for companies and an increased workload for occupational health physicians. The increased workload for COVID-19 vaccinations resulted in a delay or cancellation of certain occupational health activities, mainly including preventive medical examinations. A recent study addressing general practitioners’ perspectives on the impact of COVID-19 on preventative care showed similar results [39]. General practitioners reduced their provision of preventative care in primary care, and were also confronted with increased workload during the COVID-19 pandemic [39]. Although this was an unusual and special situation in the spring and summer of 2021, it is important that occupational health personnel can fully perform their tasks in occupational health prevention despite their participation in campaigns for generally recommended vaccinations. Despite the increased workload and the associated negative effects, three members of the occupational health personnel were convinced that there were no disadvantages of COVID-19 vaccinations in companies in general.

### 4.2. Assessment of the Pilot COVID-19 Vaccination Program in Companies

Overall, the implemented pilot COVID-19 workplace vaccination program was received positively in the German companies participating in the pilot project. First, there was good participation in COVID-19 vaccination by employees in two companies. More than half of the employees received both the first and second vaccinations in their companies. The vaccination rate of more than 90% fully vaccinated persons in our study was clearly higher compared to the vaccination rate of around 68% of the general German population at that time [40].

In addition to the findings of the employee survey, interview partners from several companies observed a relatively high willingness of employees to be vaccinated and, thus, high vaccination rates. This is perhaps due to the known positive effect of the workplace setting with regard to health-related preventive offers [41,42,43]. Due to vaccination prioritizations, the vaccination rates in Germany were relatively low in the spring of 2021 [7]. Nevertheless, the observed high willingness of employees indicates an apparently good response to the COVID-19 vaccination offer in our sample compared with a non-representative cross-sectional German study conducted at the same time [44]. It also became evident that occupational health services were perceived differently and with more appreciation by employees—as help in an “emergency situation”. During the COVID-19 pandemic, the importance of occupational health expertise and consultation for every workplace became very clear [45]. Occupational health physicians contributed significantly to the implementation of protective measures and COVID-19 vaccinations at workplaces [45]. Therefore, occupational health services played and continue to play an important role during the COVID-19 pandemic, not only for employees but also for employers [9,46]. The knowledge of and the experience gained by occupational health physicians should therefore be regarded as an important resource for future vaccination campaigns in workplaces. Occupational health physicians were an important pillar of COVID-19 vaccination campaigns in workplaces. The German study “Companies in the COVID-19 crisis” (“Betriebe in der COVID-19 Krise”) by the Institute for Employment Research showed that, in August 2021, 32% of German companies already provided COVID-19 vaccination offers to their employees through occupational health services [47]. Occupational health physicians can crucially reduce the workload of general practices in vaccination campaigns and strengthen a company as an additional place for healthcare, disease prevention, and health promotion. Further studies are necessary to examine the changed understanding of the role of occupational health services in the COVID-19 pandemic in addition to other healthcare providers. This can certainly help to improve the status of occupational health services in companies.

Besides the quite positive response, in two companies (branch of industry: segment of business of medical technologies and devices), there were some reports of a more negatively perceived response and low vaccination rates among some groups of employees. In particular, employees with an immigrant background or a low level of education were hard to reach. A recent study investigated the acceptance of COVID-19 vaccines among migrants in Germany [34]. Free and easy access to health services, the absence of language barriers, and access to correct information about vaccines played an important role [34]. The study identified that the absence of financial barriers, short waiting times, and the presence of a vaccination center nearby were relevant key factors in the decision to be vaccinated [34]. Another study identified barriers to and facilitators of vaccine uptake among migrants in the EU, the European Economic Area, the UK, and Switzerland, and highlighted the need for tailored, culturally sensitive, and evidence-based strategies to address possible acceptance barriers among migrants [48]. Another study found that lower education levels were an important predictor of hesitancy or rejection of vaccines, along with other factors such as conspiracy thinking [49].

Our study did not investigate in detail why certain groups did not want to be vaccinated even though vaccination offers were available. Thus, for future vaccination campaigns, it is important to learn more about vaccination hesitancy in workplaces and the extent to which occupational health services can be involved in increasing vaccination rates. Results from the German COSMO (COVID-19 Snapshot Monitoring) study suggest that vaccination concerns should be fully addressed, and information and educational campaigns for potential vaccine recipients should be conducted early to increase vaccination rates [50]. The interview partners addressed the need for improved vaccination information for specific employee groups, especially to address previous experiences, language barriers, and cultural backgrounds. Experience from workplace vaccination campaigns that have already been successfully implemented could also be helpful and should be incorporated into future workplace vaccination programs, for example, experiences from annual influenza vaccination campaigns [51] or other immunization campaigns [52].

### 4.3. Limitations and Strengths

The study reported here comprised both limitations and strengths. One major limitation lies within the small sample: only five companies within the pilot project agreed to take part in the study. Additionally, only two companies participated in the standardized survey with employees. Therefore, no claim to representativeness could be made here, and we cannot assume that all perspectives of the occupational health personnel and employees of each company were included. One reason for the difficult recruitment was certainly the vaccination and booster campaign running in parallel in Germany in winter 2021. Companies in Germany at this time were confronted with high infection and disease rates among their employees and were under pressure to quickly organize and provide a high number of COVID-19 vaccinations. This could have had an impact on companies’ willingness to participate in our study. In addition, a selection bias may be likely and it is assumable that companies which gained mainly positive experiences during their COVID-19 vaccination campaign were more motivated to take part in our study than those with (multiple) problems implementing their vaccination campaign. We also assume that a much more pro-vaccination population works in these critical infrastructure companies, and that participating companies had already a good workplace health management system in place. Therefore, the predominantly positive attitudes toward vaccination are not entirely surprising and, of course, are represented by the interview partners, more than half of whom had a professional background in healthcare. Another limitation may be the retrospective and summative character of the mixed methods study conducted. The pilot COVID-19 workplace vaccination program started in spring 2021, but for organizational reasons the employee surveys and qualitative interviews could not be carried out until winter 2021. Therefore, compared with a formative assessment of the pilot vaccination campaign, our results could not be used to guide the implementation of the program. Furthermore, the fact that the data collection was conducted retrospectively six months after the start of the COVID-19 vaccination campaign in companies may have changed the attitudes of the respondents. There is also the possibility of a response bias concerning the quantitative data. We assume that mainly employees with positive attitudes toward receiving COVID-19 vaccination in companies and with only few temporary contracts participated in the study. Therefore, the survey results may not be representative for the participating companies and other companies. Another limitation can be seen in the research design of our study. We conducted a study with a primarily qualitative focus enhanced by descriptive survey data. A more in-depth statistical analysis of the survey data was not the goal and was not suitable to answer our research questions. However, in the field of health services research, an interdisciplinary research field, a mixed methods study design is often used, for example, by focusing more on either the qualitative or the quantitative data [53,54].

Besides these limitations, the mixed methods design of our study was appropriate and enabled a comprehensive assessment of the pilot COVID-19 workplace vaccination program. Due to the quantitative and qualitative approaches and the triangulation of these data, it was possible to gather different perspectives including those of employees, occupational health professionals, and other key staff concerned with the management of the pilot vaccination program. The different data complemented each other, allowing for a more in-depth understanding of attitudes toward COVID-19 vaccination in workplaces in Germany. Therefore, the results may inform future vaccination campaigns in the workplace setting. The emphasis on qualitative studies is increasingly used in implementation studies and helps to “answer questions beyond effectiveness” [55]. Findings are therefore highly relevant for policy makers, who should consider the occupational setting whenever it is of the highest interest to provide easy-accessible, flexible, and time-efficient healthcare offers for employees.

## 5. Conclusions

We performed a comprehensive mixed methods study to retrospectively evaluate the pilot COVID-19 workplace vaccination program in different companies initiated by the Ministry of Social Affairs, Health and Integration in Baden-Württemberg, Germany. In summary, we identified a predominantly positive assessment of and considerable participation in the pilot COVID-19 workplace vaccination program in five companies in Southern Germany (Baden-Württemberg). We found that adding the workplace setting and including occupational health physicians could actively support the COVID-19 vaccination campaign rollout in Germany. One explanation here is the long and trusting relationship employees have with occupational health physicians, and the important role occupational health physicians incorporate in providing healthcare and prevention the workplace setting. However, we also identified challenging aspects including low vaccination rates in some companies and a generally high workload for occupational health services, especially during the overall roll-out phase of the pilot COVID-19 workplace vaccination program. Thus, we need to learn more about vaccination hesitancy in the workplace and determine the extent to which occupational health services can help to increase vaccination rates across the workforce. From our perspective, there is a crucial need for a systematic examination and evaluation of the administration of generally recommended vaccinations in the workplace setting to assess and compare their quality and derive recommendations regarding the development and implementation of future vaccination programs. Furthermore, we detected employee recognition of the important role that occupational health physicians play in combating the COVID-19 pandemic. The occupational setting was shown to present an additional opportunity on top of general practices to provide easy-accessible, flexible, and time-saving preventive healthcare and infection prevention during the COVID-19 pandemic. Present findings from the pilot COVID-19 workplace vaccination program and the important role of occupational health services (e.g., provision of vaccination services, opportunity for consultation, and occupational health expertise) should be considered to a greater extent in future vaccination campaigns in and outside of the workplace setting.

## Figures and Tables

**Table 1 vaccines-11-01082-t001:** Characteristics of the study population (questionnaire).

Variables	Categories	n (%)
Gender (n = 643)	Male Female Diverse Not specified	352 (54.7%) 268 (41.7%) 3 (0.5%) 20 (3.1%)
Age (in years) (n = 613)	Mean (SD) Range	42.8 (12.2) 18–67
Supervisor function (n = 644)	Yes No	97 (15.1%) 547 (84.9%)
Working full-time (n = 643)	Yes No	522 (81.2%) 121 (18.8%)
Fixed-term contract (n = 642)	Yes No	24 (3.7%) 618 (96.3%)
Shift work (n = 640)	Yes No	92 (14.4%) 548 (85.6%)
German nationality (n = 642)	Yes No Not specified	609 (94.9%) 23 (3.6%) 10 (1.5%)
Average work experience (in years) (n = 602)	Mean (SD) Range	18.1 (11.9) 1–58
Average tenure with the same employer (in years) (n = 613)	Mean (SD) Range	17.7 (11.4) 1–45

**Table 2 vaccines-11-01082-t002:** Overview of the study population (qualitative interviews).

Interview Number	Gender	Professional Background	Company Sector
1	male	occupational health physician	energy supply
2	female	manager for sustainability and social responsibility	medical technology
3	male	team leader for occupational health management	medical technology
4	male	occupational health physician	for different companies
5	male	occupational health physician	medical technology
6	male	head of staff department “organization and process management, inhouse consulting”	energy supply
7	female	practice assistant	medical device production
8	female	manager for quality and sustainability	pharmaceutical production
9	female	occupational health physician	medical device production
10	male	occupational health physician	pharmaceutical production

## Data Availability

The datasets generated during and/or analyzed during the current study are not publicly available due confidentiality reasons but are available from the corresponding author on reasonable request.

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
