# Peer review of "Assessing Attitudes and Participation Regarding a Pilot COVID-19 Workplace Vaccination Program in Southern Germany Considering the Occupational Health Perspective—A Mixed Methods Study"

_vaccines, 2023, doi:10.3390/vaccines11061082_

Round 1
Reviewer 1 Report (Previous Reviewer 3)
I appreciate the incorporation of editorial suggestions and attempts to better elucidate research methods, findings and conclusions suggested by the manuscript reviewers.
I still found some spacing errors that need to be fixed but otherwise the manuscript is readable.
Author Response
Please see the attachment.

Reviewer 2 Report (New Reviewer)
The paper concerns a retrospective study of COVID-19 vaccination program which was implemented in a few companies in Southern Germany. The paper fits in a long list of papers on attitudes and opinions about COVID-19 vaccination, but not to many of them concern vaccinations carried out in companies. So, as the Authors mentioned, the obtained results may be useful for designing future vaccination campaigns in workplaces.
The paper is well written and the study as well as the obtained results are clearly presented. There are some limitations of the performed study but they are clearly indicated. The Authors related the obtained results with others known from the literature.
However, it would be good if the Authors clearly indicated what are the new findings of their study compared to those known from literature, especially those which could be useful for future vaccination campaignes.
In my opinion the paper could be considered for a possible publication after minor revision.
Author Response
Please see the attachment.

Reviewer 3 Report (New Reviewer)
Very nicely written and thought out. The authors even discussed limitations in depth, which is unusual.
Author Response
Please see the attachment.

Reviewer 4 Report (New Reviewer)
1.This paper used the mixed methods to assess retrospectively attitudes and participation of employees, occupational health personnel, and key personnel regarding the rollout of a pilot COVID-19 workplace vaccination program in five German companies in May/June 2021 in Ba-den-Württemberg (Southern Germany).The research findings have practical significance in supporting the development of fu-ture programs for the administration of generally recommended vaccination in the setting work-places in Germany.
2.It should be clearly stated what the mixed methods used in the study is in the abstract.
3.It is recommended to present the quantitative research results in a table and conduct statistical analysis to observe whether the results have statistical differences.
4.Table 1 and Table 2 respectively show the characteristics of the quantitative study and qualitative research population. It is recommended to further analyze and discuss the relationship between the research results and different populations with different characteristics, in order to clarify which populations are more supportive of the plan and its reasons, which is conducive to further improving the vaccination plan.
5.Minor editing of English language required.
Minor editing of English language required.
Author Response
Please see the attachment.

Reviewer 5 Report (New Reviewer)
This manuscript is a well written description of a pilot study in workplace vaccination programs for COVID-19. The study was conducted only in two companies but the authors claims a more larger sample in Methods and other sections. They used a mixed method, using a large sample based in forms of epidemiological inquiry, whih presented adequate data and a record of ten interviews of health personal used for some statements with copy of those interview excerpts. This is the biased part of the manuscript. The authors must include in the part of the discussion after line 643 pos the manuscript the statement that most interviews were performed on health care involved personal and not on common not health care involved persons in the workplace. Selection of interviewed people due to convenience is a recurrent bias in those studies using qualitative analysis.
Author Response
Please see the attachment.

This manuscript is a resubmission of an earlier submission. The following is a list of the peer review reports and author responses from that submission.
Round 1
Reviewer 1 Report
The article feels out of scope for this journal for two primary reasons; it is not mixed methods or quantitative at all as there is no statistical analysis and it does not provide evidence-based recommendations for policy. The title and construction of the paper is therefore misleading and incorrect. For the special edition in particular, the article does not fall in line in demonstrating how such a scheme could be implemented in the future based on lessons from the past.
In assessing the facts within the introduction, the authors note,
“Therefore, those vaccination programs in the workplace were implemented during the third Corona wave, which occurred in Germany during the spring of 2021, with high infection rates among the German Population and a shortage of vaccines. ”
I am not certain what time in spring this vaccination program was implemented, but in June 2021 there was no vaccination prioritization. In other contexts and countries, front line workers were prioritized due to their elevated risks, including those in the food/delivery industry but these sites would not have clinicians or nurses present to administer vaccinations, requiring individuals to visit health institutions to be vaccinated. Taking clinicians/nurses/trained staff off site to different companies would have represented a gargantuan administrative effort to ensure adequate infection control.
“Since May 2021, more than 12,000 employees in Baden-Württemberg have been vaccinated in these companies, and initial experiences have been evaluated for use in the ongoing development of the national COVID-19 vaccination strategy.”
The vaccination of 12,000 employees relative to the 60+ million that were vaccinated by December 2021, also pales the findings of this study significantly, and demonstrates the relative failure of this program as the number of front line workers should roughly equal 10% of the total population (when examining US statistics of 31.7 mill front line workers in 2020 versus their 330 million population).
Therefore the aim of this study to “enhance development of future occupational vaccination programs, their quality and their acceptance within companies” does not seem realistic or feasible. A critique or perhaps policy piece, based on qualitative evidence, on how such a workplace vaccination scheme could be utilised would be of significantly greater use to the audience.
Only four companies participated in the qualitative interview, and two in the survey. Out of 652 employees, 548 did not work in shifts and have long-term contracts, leaving a huge population of temporary or contract workers which are not adequately represented and were a huge driving in the provision of delivery services in the pandemic. Also, the amount of professional work experience being 18 years leaves out a large number of young workers, of which make up a huge proportion of the temporary or contract workers. The selection bias is therefore very high, leading the findings to be representative of only a very small group of individuals.
I understand GRAMMS was utilised but was Cronbach’s Alpha utilised? This made me realise that there is no statistical analysis of the survey and instead verbose discussion which does not deconstruct the findings of the survey. The displaying of proportions or numbers does not qualify as a mixed methods study.
Reviewer 2 Report
Very interesting study; the author presents it by applying combined research methods and well presented both quantitative data and qualitative data. So I agree to this article for publication. This paper has also explained the limitations of future studies and research, using the latest references.
Reviewer 3 Report
You use a few conventions that I am not accustomed to. Within the manuscript, you use the initials of the investigator who was responsible for a particular aspect of the manuscript, like conducting a survey. E.R. and A.N. I will leave it to the editors but I do not think that is necessary unless you are trying to convince the editors of the contribution of each investigator to the study. It made it somewhat difficult to read.
Low-Threshold. I am not sure what that means. I am assuming that it is related to the fact that the programs made vaccines easily accessible, I think that would be the better language to use than low-threshold. If this is not what you are referring to then I believe that this needs to be explained.
While this is not possible to correct at this time, you did something unusual with your Likert scales. In one question, 1 was the positive result but in the next question 5 was the positive result. While sometimes surveyors attempt to make sure that those taking the survey are reading the question throughly, I am concerned that you may have inaccurate answers. I would recommend sticking to one convention during your next study.
You note in your limitations that you have a Selection Bias, low participation, and Recall bias, retrospective recovery of data. I think you may have another bias that you need to discuss. It is part of your selection bias. The program was predominantly offered to companies that were considered "essential" during the early phase of the pandemic. The industries that were asked to participate were likely very different from all blue-collar industries in Germany. They tended to be involved in areas such as medicine and food supply chain. Thus, I believe that you have a much more willing population in these settings which was a selection bias that occurred even before your possible selection bias towards companies that were likely to participate because the program went well. Furthermore, you had one industry that was not very successful. You may want to reveal which economic sector that industry was in as that may be an explanation that heavy industrial factory workers may be less likely to accept these interventions.
I like your emphasis that Occupational Health is likely an underutilized resource for these types of medical uses. It sounds like Occupational Health might be a bit more robust in Germany than the US but the mobilization of this resource for subsequent health emergencies should be tapped by all countries as they often have that relationship with people even when they may not have had additional primary care resources. I also thought it was very good how you were able to note that patient education seemed to be the key to success. You found that early information programs assisted with successful immunization programs and that barriers (language and education) need to be overcome in order to maximize participation.
